# Structures of two aptamers with differing ligand specificity reveal ruggedness in the functional landscape of RNA

Andrew John Knappenberger[1,2†], Caroline Wetherington Reiss[1,2†], Scott A Strobel[1,2*]

[1]Department of Molecular Biophysics and Biochemistry, Yale University, New Haven, United States; [2]Chemical Biology Institute, Yale University, West Haven, United States

**Abstract** Two classes of riboswitches related to the *ykkC* guanidine-I riboswitch bind phosphoribosyl pyrophosphate (PRPP) and guanosine tetraphosphate (ppGpp). Here we report the co-crystal structure of the PRPP aptamer and its ligand. We also report the structure of the G96A point mutant that prefers ppGpp over PRPP with a dramatic 40,000-fold switch in specificity. The ends of the aptamer form a helix that is not present in the guanidine aptamer and is involved in the expression platform. In the mutant, the base of ppGpp replaces G96 in three-dimensional space. This disrupts the S-turn, which is a primary structural feature of the *ykkC* RNA motif. These dramatic differences in ligand specificity are achieved with minimal mutations. *ykkC* aptamers are therefore a prime example of an RNA fold with a rugged fitness landscape. The ease with which the *ykkC* aptamer acquires new specificity represents a striking case of evolvability in RNA.
DOI: https://doi.org/10.7554/eLife.36381.001

**\*For correspondence:**
scott.strobel@yale.edu

†These authors contributed equally to this work

**Competing interests:** The authors declare that no competing interests exist.

## Introduction

RNA has diverse functional capabilities, which has driven speculation that the first organisms may have been RNA-based (*Breaker, 2012*; *Crick, 1968*; *Gilbert, 1986*; *Orgel, 2004*; *1968*; *Strobel, 2001*; *Woese et al., 1966*). For this hypothesis to be plausible, RNA must be adaptable; that is, capable of acquiring new functions through mutation. In the field of evolutionary biology, this trait is described as evolvability (*Kirschner and Gerhart, 1998*; *Wagner and Altenberg, 1996*). Evolvability is the propensity of a system to produce a mutated genotype that yields a beneficial phenotype under new selective pressures (*Ancel and Fontana, 2000*; *Kirschner and Gerhart, 1998*; *Wagner, 2008*; *Wagner and Altenberg, 1996*). Often, this occurs through mutation of an existing gene through divergent evolution. For example, bacterial β-lactamases demonstrate significant evolvability through mutations in the Ω-loop. This loop determines substrate specificity, but mutation or outright deletion of the loop does not dramatically affect the overall structure of the protein (*Banerjee et al., 1998*; *Hujer et al., 2001*; *Kurokawa et al., 2000*; *Wachino et al., 2004*). This locus of evolvability allows the protein to adapt to the selective pressures of novel antibiotics. The concept of evolvability has also been studied in RNA, including a notable paper by Draghi et al. (*Draghi et al., 2010*). This study found that adaptation rate is hastened when the build-up of some phenotypically neutral mutations occurs and the web of accessible phenotypes becomes broad. The speed at which an organism adapts is determined by this, as well as the ruggedness of the fitness landscape, which is related to the number of mutations required to reach a new fitness maximum.

Variant riboswitches yield insight into the evolvability of RNA. Riboswitch variants are naturally occurring riboswitches with a conserved overall fold but altered ligand specificity. Examples include the guanine/adenine riboswitches and the cyclic-di-GMP/cyclic-GMP-AMP riboswitches

**eLife digest** DNA's iconic double helix has made it possibly the most widely recognized biological molecule. The closely related RNA, however, is less well known but just as vital. In contrast with DNA's typical rigid structure, RNA is more flexible and can fold into a wide range of shapes; this allows RNA molecules to have many jobs.

Some RNA molecules form structures called riboswitches. As the name suggests, these act as molecular switches that help cells to respond to the presence of important small molecules. When a riboswitch encounters the right molecule, it changes shape, which in turn changes how the cell behaves. It is very difficult, if not impossible, to predict how a riboswitch recognizes its preferred small molecule. To address this, scientists use a technique called X-ray crystallography to directly examine the riboswitch's structure.

Knappenberger, Reiss and Strobel have now determined the structures of two recently discovered riboswitches. The two switches detect molecules called PRPP and ppGpp, respectively. These riboswitches are structurally similar to one that binds to a very different type of chemical called guanidine. The aim was to understand how similar switches respond to different signals. The results reveal that a PRPP riboswitch could become a ppGpp riboswitch just by making a single change to the RNA sequence.

Many scientists believe RNA preceded DNA and proteins in some of the earliest organisms on Earth. Understanding how RNAs have evolved and diversified could thus help to understand how early life developed. The results may also help to design synthetic riboswitches for a variety of uses. Since many riboswitches are unique to bacteria, this work could also contribute to the search for new antibiotics.

DOI: https://doi.org/10.7554/eLife.36381.002

(*Kellenberger et al., 2015*; *Mandal et al., 2003*; *Mandal and Breaker, 2004*; *Ren et al., 2015*; *Serganov et al., 2004*; *Smith et al., 2009*; *Sudarsan et al., 2008*). Bioinformatic and structural studies of the guanine/adenine and cyclic-di-GMP/cyclic-GMP-AMP aptamers showed that the altered specificity occurs simply by changing base pairing between the RNA and ligand. Recently, the *ykkC* RNA motif was identified as binding to multiple, chemically dissimilar ligands, which makes this specific scaffold a compelling target for structural studies of RNA evolvability (*Sherlock et al., 2018a*; *2018b*; *Nelson et al., 2017*).

The *ykkC* RNA was discovered in 2004 and its ligand(s) remained unknown for over a decade (*Barrick et al., 2004*; *Nelson et al., 2017*). In 2017, Nelson et al. published two pivotal discoveries regarding this motif: (1) the majority of these RNAs bind specifically to the guanidinium cation and (2) the *ykkC* riboswitch class can be divided into at least two subtypes. Subtype 1, which has approximately 1500 known examples, is the major class now known as the guanidine-I riboswitch. Subtype 2 was defined as all variants of this motif that do not recognize guanidine. The subtype 2 variants are overall quite similar to guanidine-I riboswitches. They retain the same overall fold, but possess a few characteristic differences at nucleotides crucial for guanidine binding. Notably, most of these differences are centered around a classic S-turn motif that forms the binding pocket of the guanidine-I riboswitch. A similar overall architecture with key differences in binding pocket nucleotides is a signature characteristic of a riboswitch variant (*Weinberg et al., 2017*).

Variant *ykkC* RNAs are found upstream of a variety of genes, although two major groups are apparent. One major group regulates amino acid synthesis and transport genes, which are upregulated during the stringent response. The other regulates de novo purine biosynthesis, which produces purine nucleotides from smaller metabolites under conditions where intact nucleobases are not available (*Sherlock et al., 2018a*; *2018b*; *Ebbole and Zalkin, 1987*; *1989*). These riboswitches were designated as *ykkC* subtype 2a and 2b, respectively. When compared to guanidine riboswitches, subtypes 2a and 2b harbor systematic changes to residues directly involved in guanidine binding, which led to the suggestion that they may have different ligand specificity. For example, where guanidine riboswitches have a conserved adenosine residue (A46 in the guanidine-I structure solved by Reiss et al.), subtypes 2a and 2b have a pyrimidine (C49 in the present study) (*Battaglia et al., 2017*; *Reiss et al., 2017*). Sorting the entire *ykkC* class by the identity of this position alone results in a

strikingly complete segregation of guanidine-related gene contexts from those that are incongruent with mitigation of guanidine toxicity (*Nelson et al., 2017*). Alignment of subtype 1, 2a, and 2b sequences also shows an extension of conservation at both the 5′ and 3′ ends of the 2a and 2b aptamer subtypes. These key differences in conserved residues and gene contexts suggested that these *ykkC* variants have altered ligand specificity while retaining the same overall architecture.

Subtype 2a and 2b *ykkC* riboswitches do not retain the ligand specificity of their parent riboswitch. Using transcription termination and in-line probing assays, Sherlock et al. found that neither subtype is responsive to guanidine (*Sherlock et al., 2018a*; *2018b*). Instead, subtype 2a is responsive to guanosine tetra/pentaphosphate ((p)ppGpp, hereafter referred to as ppGpp), an alarmone that regulates the stringent response (*Cashel and Gallant, 1969*; *Dalebroux and Swanson, 2012*; *Gaca et al., 2015*). Subtype 2b is responsive to phosphoribosyl pyrophosphate (PRPP), a precursor in purine biosynthesis. Like the guanidine riboswitch, both function as ON switches. The consensus motifs for subtypes 2a and 2b are remarkably similar to each other, even relative to other *ykkC* RNAs (*Sherlock et al., 2018a*; *2018b*). The most apparent difference is a highly-conserved guanosine (G96 in this study) in subtype 2b that is not conserved in subtype 2a. This residue is equivalent to G89 in the guanidine-I riboswitch and is a conserved part of its S-turn motif. Although bioinformatic data suggest that variation in G96 is central to the structural differences between subtype 2a and 2b riboswitches, its precise role in this context remains uncertain.

Unlike guanidine, the biological roles of PRPP and ppGpp are both well-documented. PRPP is an activated form of ribose 5-phosphate, and a major macromolecular building block (*Hove-Jensen et al., 2017*). It is a central metabolite used in biosynthesis of purine and pyrimidine nucleotides, the amino acids histidine and tryptophan, nicotinamide adenine dinucleotide, thiamine diphosphate, flavins, and pterins (*Hove-Jensen, 1988*; *Jiménez et al., 2008*; *White, 1996*). The centrality of PRPP within metabolism makes it an appealing target for regulation. ppGpp is an alarmone that initiates the stringent response, a global reaction to nutrient starvation in bacteria (*Cashel and Gallant, 1969*; *Cashel and Kalbacher, 1970*; *O'Farrell, 1978*; *Potrykus and Cashel, 2008*). Amino acid starvation triggers synthesis of ppGpp and it binds to a variety of effector molecules to initiate sweeping changes in the cell's transcriptional profile, including a reduction in tRNA and rRNA synthesis and an increase in transcription of amino acid biosynthesis genes (*Cashel, 1970*; *Paul et al., 2005*; *Ryals et al., 1982*; *van Ooyen et al., 1976*). Consistent with a role in the stringent response, the ppGpp riboswitch turns on transcription of amino acid biosynthesis and transport genes in response to alarmone binding.

Although the tree topology is unknown, a common ancestral RNA likely diverged to recognize guanidine, PRPP, and ppGpp in spite of the chemical and structural diversity among these ligands. PRPP and ppGpp are more similar to each other than either is to guanidine, which reflects the greater similarity in their aptamers. While guanidine harbors a single delocalized positive charge, PRPP and ppGpp harbor multiple separate loci of negative charge. Guanidine is small and achiral with three-fold rotational symmetry, while PRPP and ppGpp are larger, chiral, asymmetric molecules. PRPP and ppGpp both contain ribose sugars and pyrophosphate moieties, but ppGpp has an entire guanine base that PRPP lacks. Bioinformatic evidence suggests that the 2a and 2b aptamers represent an especially concise solution to a central biophysical problem: biologically relevant switching entails recognition of a cognate ligand and rejection of structurally similar alternatives.

We set out to determine how three RNA elements with a common scaffold could recognize such dissimilar ligands with high specificity. Central questions include how a polyanionic macromolecule differentially recognizes two distinct small polyanions, and how the presence or absence of the guanine base changes the RNA's recognition strategy. To address these questions of molecular recognition by RNA, we report the near-atomic resolution structure of a native *ykkC* 2b riboswitch in complex with PRPP via X-ray crystallography. We also convert this construct into a ppGpp aptamer with a single G96A mutation and present the structure of the mutant bound to ppGpp. This structural and biochemical information reveals how the *ykkC* RNA differentiates between ppGpp and PRPP. This study showcases the functional plasticity of RNAs and the evolvability of RNA function from a single structural scaffold.

## Results

### The structure of the wild-type PRPP aptamer and a single point mutant ppGpp aptamer

To understand the basis of ligand recognition by the PRPP riboswitch, we determined the crystal structure of the aptamer domain of the *ykkC* 2b riboswitch from *Thermoanaerobacter mathranii* at 2.5 Å resolution in the presence of its native ligand, PRPP (*Supplementary file 1*). PRPP is an activated metabolic intermediate. As a result, it is highly unstable. It degrades on a time course of minutes to hours via several mechanisms in the presence of divalent metal ions, acidic or basic pH, and/or elevated temperatures (*Dennis et al., 2000*; *Hove-Jensen et al., 2017*; *Khorana et al., 1958*; *1955*; *Meola et al., 2003*; *Remy et al., 1955*). However, binding to the PRPP riboswitch aptamer domain protects PRPP on a time scale of hours to days (*Figure 1—figure supplement 1*). The stabilizing effect of the aptamer permitted crystals of the intact complex to be observed after two days. Once formed, unfrozen crystals disappeared after approximately five to ten days, underscoring the need for prompt crystallization and cryogenic preservation in this study. The structure was solved by molecular replacement using the guanidine-I aptamer as an initial model. After model building and refinement, the model fit the data with an $R_{work}$ of 0.216 and an $R_{free}$ of 0.253.

Like its parent aptamer, the PRPP riboswitch contains two adjacent helical stacks (*Figure 1*). P3 forms a large portion of the binding pocket, and a conserved loop at the end of P3 docks into P1a. This allows conserved nucleotides from P1a to participate in ligand recognition. P1, P1a, P1b, and P2 together form a continuous coaxial stack adjacent to P3. However, unlike the guanidine aptamer, the PRPP aptamer has structured tails at the 5′ and 3′ ends that are not conserved in the guanidine riboswitch. The ends pair to form an additional short helix that we have termed P0, resulting in a four-way junction between P0, P1, P2, and P3. P0 coaxially stacks with P3 and extends the binding pocket for recognition of the larger PRPP ligand. The overall architecture of the PRPP aptamer reveals that it is a rather conservative adaptation of the guanidine aptamer with key differences that allow for PRPP recognition.

Although PRPP is unstable in solution, it has high occupancy in this crystal structure. PRPP is modeled with an occupancy of 1, and its B factors refined similarly to those of nearby residues. The quality of the fit between the electron density data and this model shows that a combination of protection by the riboswitch and a vast molar excess of ligand permitted a high degree of aptamer saturation when data were collected.

PRPP is a potentially challenging ligand for RNA to recognize; it has three negatively charged phosphate groups and lacks a moiety resembling a nucleobase. PRPP is known to interact with two divalent metal ions per molecule in solution. The 5-phosphate associates weakly with one metal and the pyrophosphate moiety more strongly coordinates a second metal (*Thompson et al., 1978*). In the current model, these two metals are present in the complex with the riboswitch (*Figure 2*). One metal (M1) associates with the 5-phosphate, and the second metal (M2) associates with the pyrophosphate. Both metals form contacts bridging PRPP and the RNA aptamer. A third metal ion, M3, forms a water-mediated coordination to the 5-phosphate. The same water molecule also coordinates M1. The three phosphate groups are major elements of recognition via interactions with nucleobase amines and divalent metal ions.

This construct crystallizes in the presence of $BaCl_2$, so both $Ba^{2+}$ and $Mg^{2+}$ are present in the crystallization condition. M1 and M3 are modeled as $Ba^{2+}$ due to the appearance of large positive peaks in the electron density map when they are modeled as $Mg^{2+}$. M2 is modeled as $Mg^{2+}$, but exhibits coordination distances higher than expected for this species (*Figure 3—figure supplement 1*). The aptamer binds PRPP with nearly equal affinity in the presence of either $Ba^{2+}$ or $Mg^{2+}$ alone ($2.0 \pm 0.4$ and $2.0 \pm 0.3$ μM, respectively). Given that both metals support binding, we expect that there may be partial occupancy of these two species that cannot be resolved at this resolution.

The 5-phosphate of PRPP experiences recognition by a metal ion and the amino groups of conserved nucleotides (*Figure 3A*). The N1 and N2 of G48 form hydrogen bonds with two phosphate oxygens, while the N4 of C78 hydrogen bonds to the third non-bridging phosphate oxygen. The 5-phosphate also coordinates M1, which is held in place by coordination interactions with a non-bridging phosphate oxygen of C77 and the O2 of C49. The residue equivalent to C49 is conserved as an adenosine in the guanidine-I riboswitch but is a pyrimidine in PRPP and ppGpp riboswitches, and the identity of residue 49 was used as a marker to distinguish between these two variants

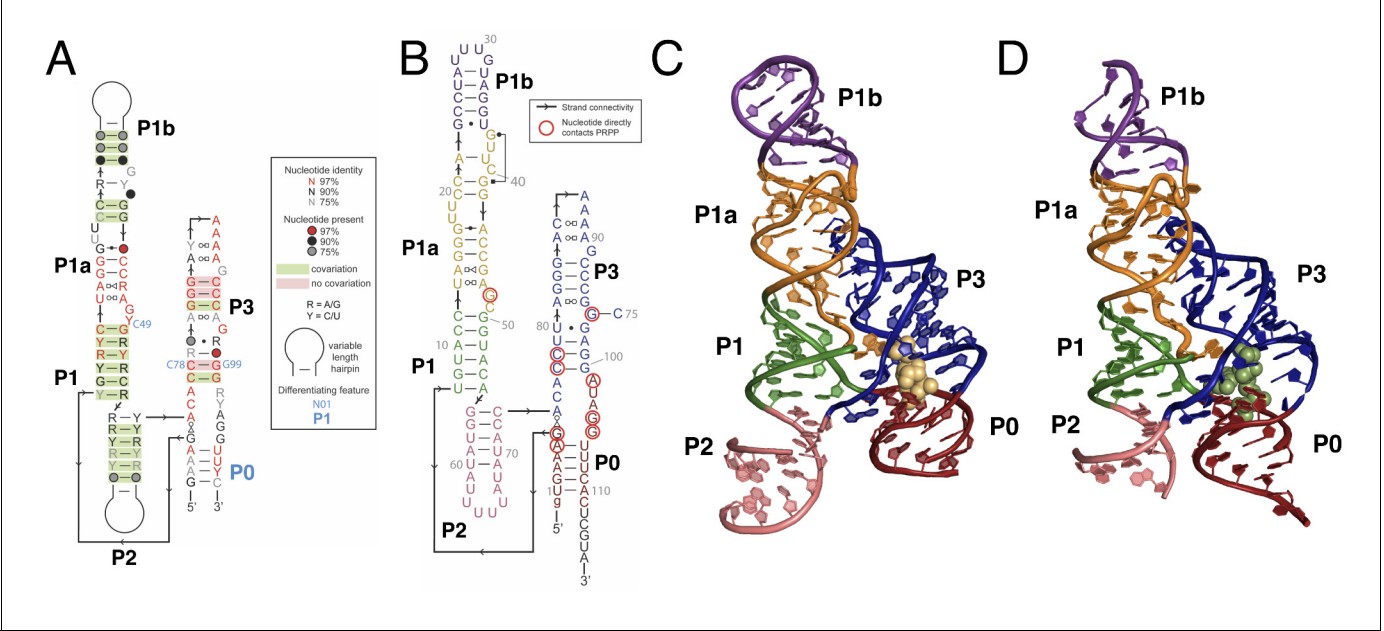

**Figure 1.** Overall structure of the PRPP riboswitch and its G96A mutant, which is a ppGpp aptamer. (**A**) Consensus sequence of the PRPP riboswitch, adapted from Sherlock et al. (**Sherlock et al., 2018b**). The secondary structure has been updated to show structural information gained from the present study. The sequence is depicted as in Sherlock et al. (see key). Nucleotides noted in blue are important bioinformatic differences between PRPP riboswitches and guanidine riboswitches. Base pair notation is as published previously (**Leontis and Westhof, 2001**). (**B**) Secondary structure of the PRPP riboswitch aptamer from T. mathranii. Nucleotides are colored by paired region. Paired regions are indicated in bold. Sequence numbering is indicated in gray. Nucleotides that directly contact PRPP are circled in red, and arrows indicate strand connectivity. (**C**) Crystal structure of the PRPP riboswitch. Chain A is shown. The RNA is depicted as a cartoon and PRPP is depicted as yellow spheres. Nucleotides are colored by paired region as in B. (**D**) Crystal structure of the G96A mutant. Chain A is shown. The RNA is depicted as a cartoon and ppGpp is depicted as green spheres. Nucleotides are colored by paired region as in B.

DOI: https://doi.org/10.7554/eLife.36381.003

The following source data and figure supplements are available for figure 1:

**Source data 1.** Summary of fitted binding data without Bmax constraints.
DOI: https://doi.org/10.7554/eLife.36381.006
**Source data 2.** Raw binding data.
DOI: https://doi.org/10.7554/eLife.36381.007
**Figure supplement 1.** Autoradiograph of a representative PAGE gel from dissociation constant determination for the PRPP aptamer.
DOI: https://doi.org/10.7554/eLife.36381.004
**Figure supplement 2.** Data from equilibrium dialysis experiments and fits used to calculate dissociation constants.
DOI: https://doi.org/10.7554/eLife.36381.005

(**Sherlock et al., 2018a**; **2018b**). The O6 of G48 coordinates M3, but M3 is too distant from the 5-phosphate to be directly coordinated by it.

The ribose moiety of PRPP also makes extensive interactions with the RNA aptamer (**Figure 3B**). The sugar edge of G96 forms hydrogen bonds with the 2- and 3-hydroxyl groups. The N4 of C77 donates a hydrogen bond to the ribose oxygen, and the N1 group of G104 donates a hydrogen bond to the 2-hydroxyl group. These three residues are all highly conserved in the consensus sequence of this aptamer. At 2.5 Å resolution, conclusive determination of the sugar pucker is not possible, but a C2-endo pucker is the most likely conformation in this complex and it fits the electron density data well. This conformation avoids a steric clash between the 2-hydroxyl and the β-phosphate and allows the 3-hydroxyl to coordinate M2. This conformation is also consistent with previously reported structures of PRPP in complex with macromolecules (**Evans et al., 2014**; **González-Segura et al., 2007**; **Héroux et al., 2000**).

The P0 region of the aptamer extends below P3 and permits a suite of interactions with the pyrophosphate group of PRPP (**Figure 3C–D**). The β-phosphate of PRPP is more extensively recognized than the α-phosphate. The O6 of G6 coordinates M2, which in turn forms several interactions with

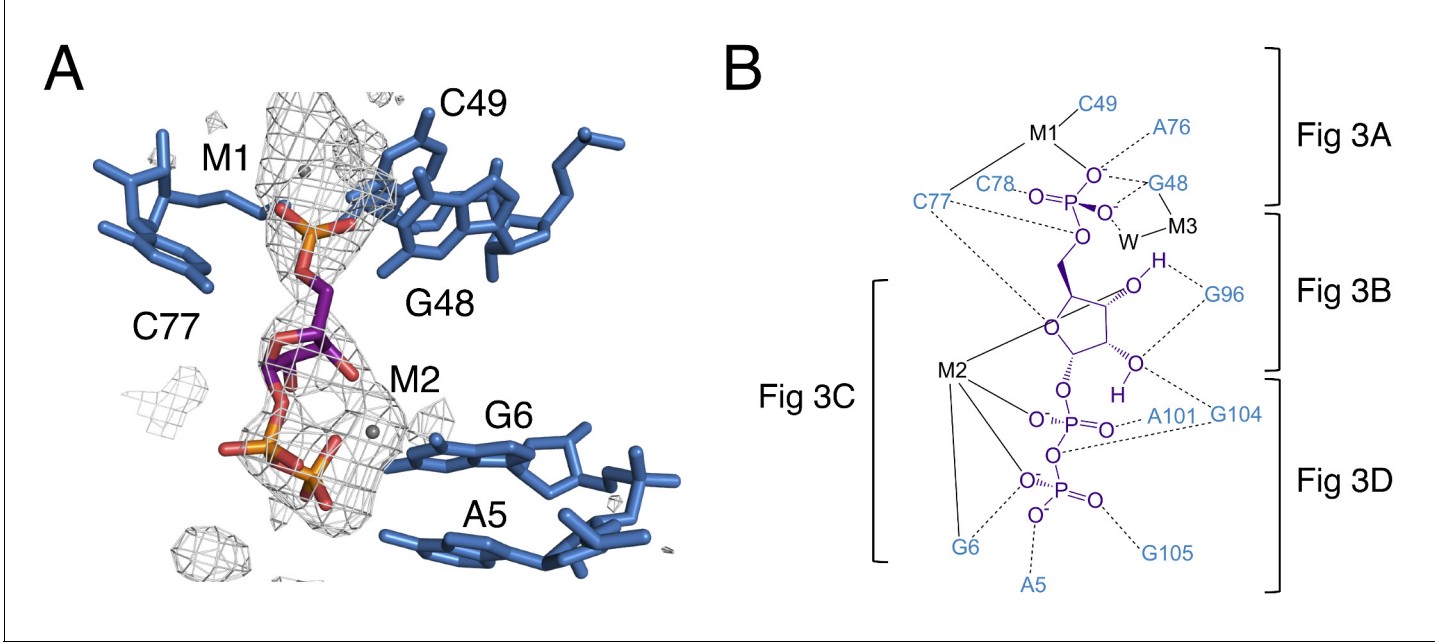

**Figure 2.** The binding pocket of the PRPP riboswitch. (A) Crystal structure of the ligand-binding site in chain A. Relative to *Figure 1*, the structure is rotated 180° about the y axis. PRPP is depicted as sticks and colored by element with purple carbons. Nucleotides are depicted as blue sticks. Metal ions are depicted as gray spheres. Individual nucleotides and metals are labeled. An $F_O$–$F_C$ map contoured at 2.5 **σ** is shown as a gray mesh. The map was calculated using an otherwise complete model lacking PRPP, M1, and M2. (B) Ligand interaction map. The map is colored essentially as in A. All RNA and metal contacts to PRPP are shown. Dashed black lines indicate hydrogen bonds. Solid black lines indicate coordination to a metal ion. Brackets indicate interactions shown in individual panels of *Figure 3*.

DOI: https://doi.org/10.7554/eLife.36381.008

the pyrophosphate group (*Figure 3C*). The N6 group of the weakly conserved A101 (>75% conserved as a purine) contacts a non-bridging oxygen of the α-phosphate (*Figure 3D*). The N6 group of A5 and the N1 groups of G6 and G105 make direct contacts with non-bridging oxygens of the β-phosphate. An abrupt deformation in the local backbone conformation positions A103 under G105, allowing a lone pair-π interaction to form between the O6 atom of G105 and the six-membered ring of A103 (*Chawla et al., 2017*; *Egli and Sarkhel, 2007*; *Ran and Hobza, 2009*; *Sarkhel and Desiraju, 2003*; *Singh and Das, 2015*). The present results show that the PRPP aptamer recognizes its ligand through a shifted and extended helical ligand-binding region, allowing for the retention of bound metal ions and extensive hydrogen bond donation to phosphate groups.

The intracellular PRPP concentration in bacteria is estimated to be in the millimolar range (*Hove-Jensen et al., 2017*; *Jendresen et al., 2011*; *Jensen et al., 1979*; *Nygaard and Smith, 1993*; *Saxild and Nygaard, 1991*; *Schneider and Gourse, 2004*; *Yaginuma et al., 2015*). However, enzymes and protein regulatory elements that sense PRPP concentrations in bacteria typically have micromolar dissociation ($K_d$) or Michaelis ($K_M$) constants (*Bera et al., 2003*; *Hove-Jensen et al., 2017*; *Jørgensen et al., 2008*). Sherlock and colleagues recently found that the $T_{50}$ (the ligand concentration that produces half-maximal effect) of a PRPP riboswitch in transcription termination assays is 90 μM (*Sherlock et al., 2018b*). We determined the $K_d$ of the riboswitch aptamer domain for PRPP (*Table 1*, see also *Figure 1—figure supplement 2A*) by equilibrium dialysis using radiolabeled [β-[33]P]-PRPP. This assay yields a $K_d$ of 2.0 ± 0.3 μM. There are two notable differences between the present experimental system and that employed by Sherlock et al. First and most importantly, the present study examines binding affinity in an isolated aptamer domain, while Sherlock et al. focused on the ability of the full riboswitch to terminate transcription. The full system is governed by the kinetics of ligand association and RNA folding, while the present experimental system only measures the thermodynamics of ligand binding. Also, in this study, [β-[33]P]-PRPP was used in trace quantities and the amount of intact PRPP remaining in each sample was carefully measured to deconvolute the counts obtained from intact PRPP and the counts obtained from breakdown products. Sherlock

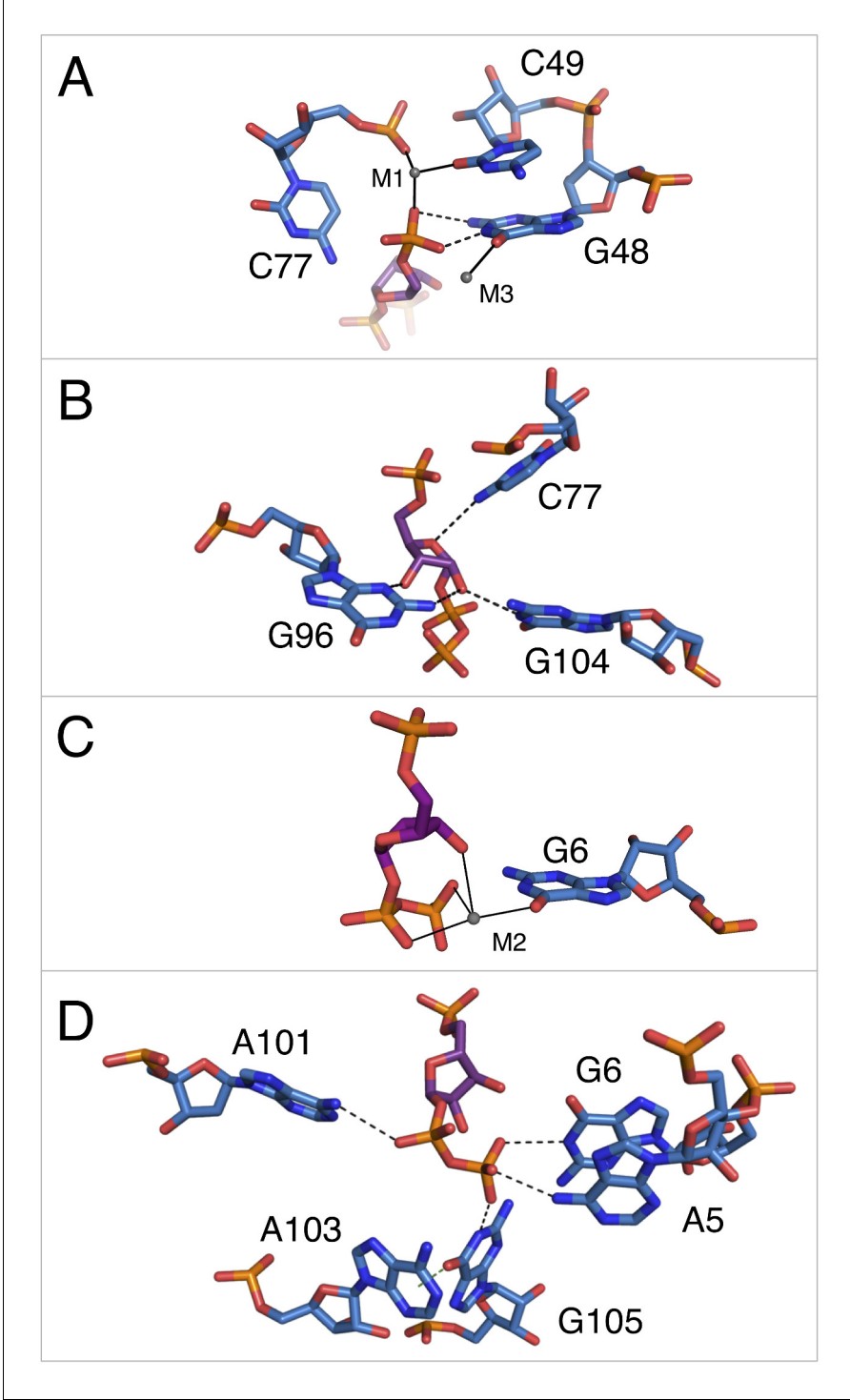

**Figure 3.** Notable contacts to the PRPP ligand in chain A. PRPP is depicted as sticks and colored by element with purple carbons. Nucleotides are depicted as sticks and colored by element with blue carbons. Individual nucleotides and metals are labeled. Dashed black lines indicate hydrogen bonds. A dashed green line shows the lone pair-π interaction between A103 and G105. Solid black lines indicate coordination to a metal ion. Relative to *Figure 1*, the structure is rotated 180° about the y axis. Panel A is additionally rotated nearly 90° about the x axis. Panel D is rotated approximately 45° about the x axis in the opposite direction. (**A**) Contacts among the 5-phosphate of PRPP, residues G48, C49, C77, and metal ions M1 and M3. (**B**) Hydrogen bonds between the ribose of PRPP and residues C77, G96, and G104. (**C**) Coordination of metal M2 by PRPP and residue G6. (**D**) Recognition of the pyrophosphate group of PRPP by residues A5, G6, A101, and G105.

*Figure 3 continued on next page*

*Figure 3 continued*

DOI: https://doi.org/10.7554/eLife.36381.009

The following figure supplement is available for figure 3:

**Figure supplement 1.** Coordination of metals by chain A of the PRPP aptamer.

DOI: https://doi.org/10.7554/eLife.36381.010

et al. used unlabeled PRPP and could not quantify the extent of degradation, likely resulting in some underestimation of PRPP's ability to terminate transcription. The present data show that the affinity of the complex is at least of low micromolar affinity, placing it well within the range observed for complexes of PRPP with protein elements (*Bera et al., 2003*; *Jørgensen et al., 2008*).

In parallel with structural inquiries into the PRPP riboswitch, crystallization of native ppGpp aptamers was pursued. However, crystallization was unsuccessful with the subset of ppGpp aptamers tested. Considering the evident versatility of the *ykkC* motif and the overt similarity between the consensus sequences of *ykkC* RNA subtypes 2a and 2b, a specificity switch of the PRPP aptamer to a ppGpp aptamer was pursued via mutation as an alternative strategy.

Close examination of the consensus motifs of the PRPP and ppGpp riboswitch aptamers revealed that the ppGpp aptamer consensus sequence was almost entirely a subset of the PRPP aptamer consensus sequence, with the PRPP aptamer generally having more stringent requirements than the ppGpp aptamer. The most salient difference between the two consensus sequences is at position 96. In the PRPP aptamer, this position is >97% conserved as a guanosine, but this conservation is lost in the ppGpp aptamer. In the ppGpp aptamer, the lack of conservation in this region complicates the process of sequence alignment. However, it appears that this nucleotide is not always present and, when it is, it appears to be conserved as A, C or U, but not G (*Sherlock et al., 2018a*). The dramatic difference in conservation at this site suggested that it may be critical for differential recognition of PRPP and ppGpp.

We mutated position 96 in the *T. mathranii* PRPP aptamer from guanosine to adenosine, generating the G96A mutant. The wild-type aptamer shows low affinity for ppGpp ($K_d = 91 \pm 3$ μM) and 46-fold greater affinity for PRPP ($K_d = 2.0 \pm 0.3$ μM) (*Table 1*). Conversely, the G96A mutant binds ppGpp with an affinity equivalent to that of wild-type for PRPP ($K_d = 1.8 \pm 0.1$ μM), but PRPP binding is abolished in the mutant up to 400 μM RNA (estimated $K_d = 1600 \pm 200$ μM). The G96A mutant has approximately 900-fold higher affinity for ppGpp than PRPP. The G96A mutation thus strikingly resulted in approximately a 40,000-fold switch in ligand specificity from PRPP to ppGpp. The mutant's affinity for ppGpp is well within the range of native aptamers tested (data not shown).

## Co-crystal structure of the generated ppGpp aptamer and its ligand

Having shown that the G96A mutant is a ppGpp aptamer, we solved its crystal structure in the presence of ppGpp to 3.1 Å resolution. The crystallization conditions that reproducibly gave rise to co-crystals of the wild-type PRPP aptamer did not yield comparable results for co-crystals of the G96A mutant. However, the G96A mutant was found to crystallize in a separate condition that also produced crystals of the wild-type aptamer. The crystallization reagent used for G96A lacks barium, which was the most abundant divalent metal ion in the wild type crystallization condition. Potassium chloride, sodium chloride, and magnesium chloride were present in the crystallization drops. $K^+$ and $Mg^{2+}$ ions are observed in the mutant crystal structure. The best mutant crystal diffracted to a resolution of 3.1 Å and its structure was solved by molecular replacement using chain A of the PRPP

**Table 1.** Dissociation constants for PRPP and ppGpp binding to the wild type and G96A *T. mathranii* aptamers with calculated fold specificity changes.

**Dissociation constants for WT and G96A binding to PRPP and ppGpp**

| Construct | $K_d$ for PRPP | $K_d$ for ppGpp | Fold specificity for PRPP over ppGpp | Estimated magnitude of overall specificity switch |
|---|---|---|---|---|
| Wild type | 2.0 ± 0.3 μM | 91 ± 3 μM | 46 | ~40,000 |
| G96A | 1600 ± 200 μM | 1.8 ± 0.1 μM | ~0.001 | |

DOI: https://doi.org/10.7554/eLife.36381.011

riboswitch as an initial model. The asymmetric unit contained four aptamer molecules. Molecular replacement and refinement revealed robust density for the electron-dense pyrophosphate groups of ppGpp as well as its guanine base. In the initial solution and throughout refinement, the quality of the electron density was worse in chain D compared to chains A-C. The model of chain D is consistent with that of chains A-C, but is excluded from discussion in the text.

Overall, the architecture of the G96A mutant is very similar to that of the wild-type aptamer (*Figure 1D*). Notably, the $2F_O - F_C$ map generated directly by molecular replacement showed no electron density in the former location of the ribose and phosphate of G96. Additional lack of electron density for the ribose of G95 and the phosphate of G97 immediately suggested that the G96A mutation caused major conformational rearrangement in this region.

The orientation of the ppGpp ligand was determined by examining an $F_O - F_C$ map where the input model lacked ppGpp. The positions of the 5′ and 3′ pyrophosphate groups of ppGpp are easily inferred from the available electron density data, which clearly show that the 5′ pyrophosphate occupies the former position of the pyrophosphate of PRPP. In this orientation, there is high electron density at the phosphates and lesser electron density at the 4′ and 5′ carbons, as expected (*Figure 4A*). This results in the ppGpp ribose having the opposite orientation of the PRPP ribose. The 5′ pyrophosphate is oriented toward P0 in the ppGpp structure, but the 5-phosphate is oriented away from P0 in the PRPP structure (*Figure 1C,D*, *2A* and *4A*). Several metal ions appear to associate with the pyrophosphate moieties. These were initially assigned as magnesium ions or water molecules, and subsequently assigned as more electron dense potassium ions due to implausibly low B factors after refinement. The positioning of these entities is highly variable among the molecules in the asymmetric unit, suggesting that they do not make essential contributions to ligand recognition, but may provide general charge stabilization.

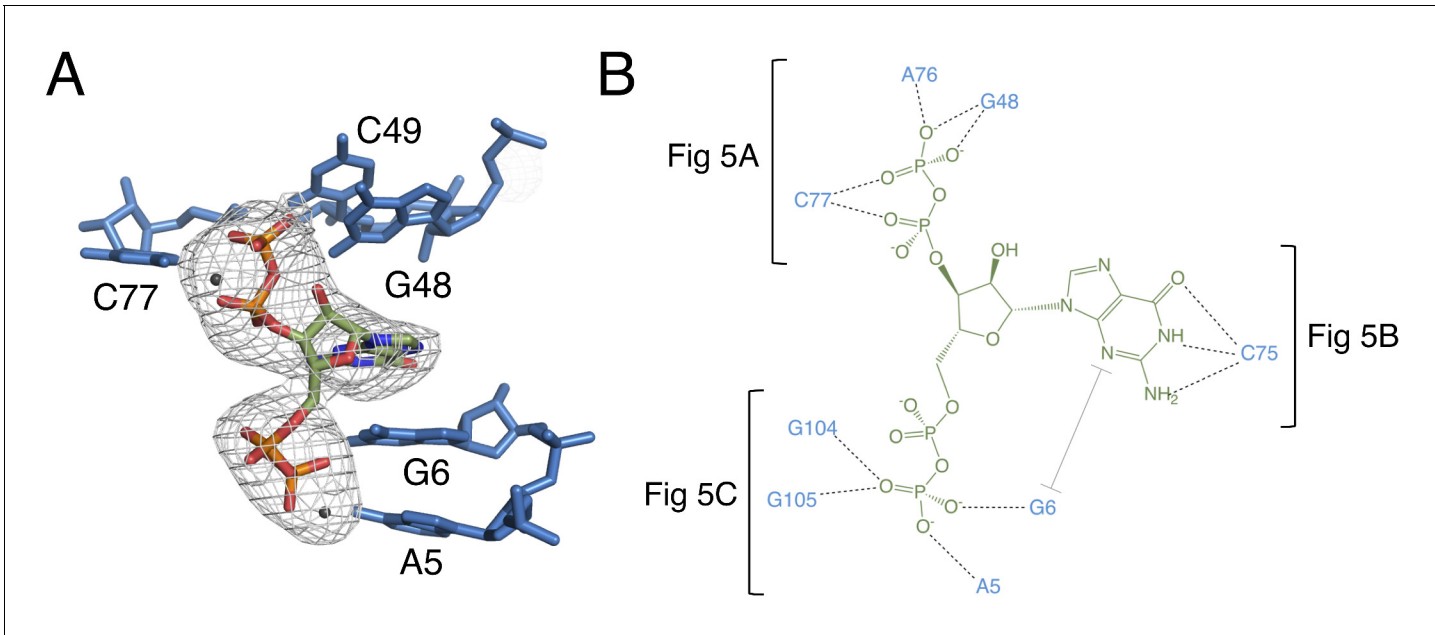

**Figure 4.** The binding pocket of the G96A mutant in complex with ppGpp. (**A**) Crystal structure of the ligand-binding site in chain A. ppGpp is depicted as sticks and colored by element with green carbons. Nucleotides are depicted as blue sticks. Metal ions are depicted as gray spheres. Individual nucleotides are labeled. An $F_O - F_C$ map contoured at 3.0 σ is shown as a gray mesh. The map was calculated using an otherwise complete model lacking ppGpp and nearby metals. Relative to *Figure 1*, the structure is rotated 180° about the y axis. (**B**) Ligand interaction map. The map is colored essentially as in A. All RNA contacts to ppGpp are shown. Dashed black lines indicate hydrogen bonds. The grey bracket indicates base stacking. Black brackets indicate interactions shown in individual panels of *Figure 5*.

DOI: https://doi.org/10.7554/eLife.36381.012

The following figure supplement is available for figure 4:

**Figure supplement 1.** A comparison of ppGpp modeled in the syn conformation and the anti conformation.

DOI: https://doi.org/10.7554/eLife.36381.013

The guanine base of ppGpp is modeled in the *syn* conformation (*Figure 4—figure supplement 1*). At 3.1 Å resolution, it is essential to inform this decision with the expected behavior of the chemical constituents in addition to the available electron density data. The shape of the electron density appears visibly more consistent with the *syn* conformation than the *anti* conformation. The chemical environment is also more plausible. In the *syn* conformation, the guanine base of the ligand forms three hydrogen bonds with C75 in a Watson-Crick base pair. In the *anti* conformation, the Hoogsteen face of the guanine base would form just one hydrogen bond with the Watson-Crick face of C75. Refinement of the ligand in the *anti* conformation created steric clashes or very short hydrogen bonds between the O6 of ppGpp and the N4 of C75, while simultaneously yielding unusually long hydrogen bonds (>3.5 Å) between the N7 of ppGpp and the N3 of C75. Modeling a Watson-Crick base pair (*syn* conformation) is consistent with a recent study showing that the equivalent of a C75U mutant in a native ppGpp riboswitch confers specificity to adenine-containing ligands over guanine-containing ligands (*Sherlock et al., 2018a*). The *syn* conformation of ppGpp was previously observed in a 2.0 Å X-ray crystal structure of an *E. coli* lysine decarboxylase, LcdI (*Kanjee et al., 2011*). Finally, a structural overlay of the wild-type and G96A structures at C75 shows that in the *syn* conformation, the base of ppGpp in the G96A structure occupies the same position as the base of G96 in the wild type structure.

The 3′ pyrophosphate of ppGpp consistently sits in a pocket lined with hydrogen bond donors (*Figure 5A*). The N4 of C77, the N1 and N2 of G48, and the 2′OH of A76 all make hydrogen bonds to the phosphate oxygens. A76 and G48 form a type I A-minor-like interaction in which the Watson-Crick edge of G48 interacts with the ligand, rather than being involved in a canonical base pair. While the position of the 5′-pyrophosphate of ppGpp is relatively invariable, the 3′-pyrophosphate occupies a slightly different position in each molecule of the asymmetric unit. Consistent with this model, the 3′ pyrophosphate atoms have slightly higher B factors than the rest of the ligand (~138 $Å^2$ for the 3′ pyrophosphate compared to ~119 $Å^2$ for the 5′ pyrophosphate). In chain A, the 3′-β-phosphate has one oxygen that accepts a hydrogen bond from the N1 of G48, a second oxygen that accepts a hydrogen bond from the N4 of C77, and a third, unrecognized oxygen. The recognition strategy is slightly different for chains B and C. While these observations may suggest genuine variation in recognition of the 3′ pyrophosphate, definitive interpretation is confounded by the comparatively lower resolution of this data set.

The guanine base of ppGpp is buried in the RNA and is the focal point of ligand recognition. In the PRPP aptamer, the highly conserved C75 forms a Watson-Crick base pair with G96. The G96A mutant ppGpp aptamer recognizes its ligand through a similar Watson-Crick base pair between the G of ppGpp and C75 (*Figure 5B*). The guanine base of the ligand is also recognized via stacking with G6 and is 56% buried, compared to 38% of ppGpp overall. Such extensive recognition of ppGpp's nucleobase suggests a likely mechanism for the mutant's observed discrimination for ppGpp over PRPP. In the native PRPP aptamer, C75 is in the same location near the binding pocket, poised to form this interaction with ppGpp. However, the highly conserved G96 is also available to form this base pair and its spatial proximity to C75 raises its effective concentration, making it potentially able to outcompete ppGpp for this base pairing interaction. This model is consistent with the observation of low-affinity ppGpp binding ($K_d$ = 91 ± 3 μM) in the wild type PRPP aptamer and explains why a single mutation at position 96 renders this aptamer capable of recognizing ppGpp with high affinity. The ribose is not recognized by the aptamer, leaving the guanine base and pyrophosphates as the major points of recognition.

Recognition of the 5′-pyrophosphate of ppGpp is extensive; its phosphate oxygens accept several hydrogen bonds from amino groups of conserved nucleobases (*Figure 5C*). The 5′-β-phosphate has three oxygen atoms that can accept hydrogen bonds from the aptamer. One of these oxygens accepts a hydrogen bond from the N6 group of A5. The second oxygen can accept hydrogen bonds from the N6 of A5 and N1 and N2 of G6, although it is not expected that these would all form simultaneously. The third oxygen can accept hydrogen bonds from the N1 and N2 of G105 and the N1 of G104. As with the previous oxygen, it is not expected that these would all form simultaneously. The 5′-α-phosphate appears to be unrecognized, consistent with its similar position to the poorly recognized α-phosphate of PRPP in the native structure.

Nucleotide A74 appears to play a conserved structural role in the PRPP and ppGpp aptamers. In all three *ykkC* subtypes it forms a noncanonical base pair with G6, which directly contacts PRPP and ppGpp, suggesting that it plays a role in positioning G6 (*Figure 6—figure supplement 1*). In the

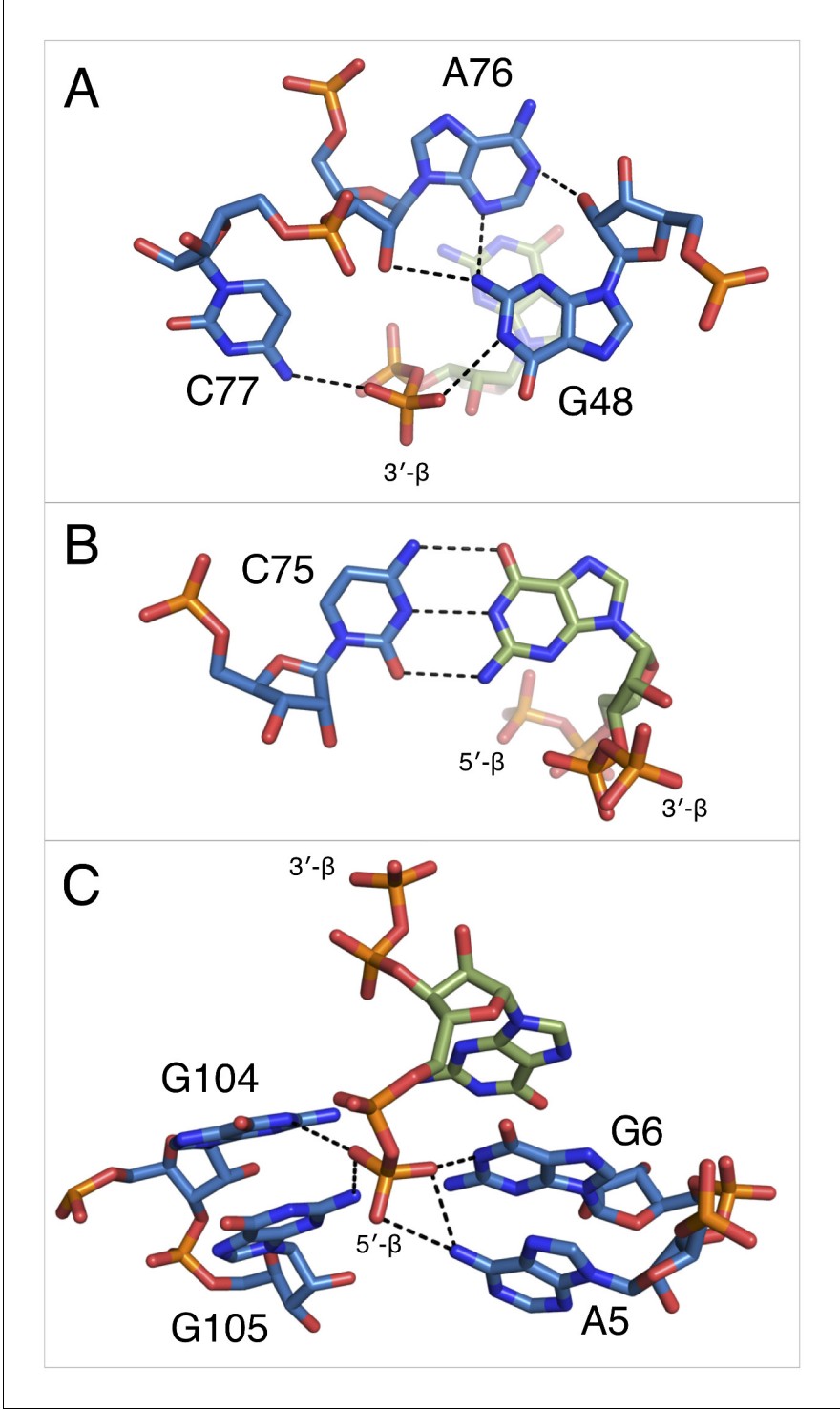

**Figure 5.** Notable contacts to the ppGpp ligand in chain A. ppGpp is depicted as sticks with green carbons and is colored by element. RNA is depicted as sticks with blue carbons and is colored by element. Dashed black lines indicate hydrogen bonds. (**A**) hydrogen bonds donated from amino groups in C77 and G48 to the 3′-β-phosphate of ppGpp, as well as the hydrogen bond network that constructs this part of the binding pocket. The 2′OH of A76 is close enough in chains B and C to form an additional hydrogen bond to the 3′-β-phosphate. (**B**) The Watson-Crick base pair between C75 in the RNA and the ppGpp ligand. (**C**) hydrogen bonds donated from amino groups in A5, G6, G104, and G105 to the 5′-β-phosphate of ppGpp.

DOI: https://doi.org/10.7554/eLife.36381.014

guanidine-I riboswitch, this nucleotide is not conserved. However, A6 in the guanidine-I crystal structure flips out to form the same non-canonical base pair with A68 (equivalent to A74) that is observed in the present study (*Battaglia et al., 2017*; *Reiss et al., 2017*). The lack of conservation at this position does not support a role in guanidine recognition, but this conserved interaction is observed in all three aptamers (*Nelson et al., 2017*).

The guanidine-I and PRPP *ykkC* aptamers each have an S-turn motif in the P3 helix. In the guanidine-I aptamer, the orientation of G88 is reversed relative to its stacking partners and G89 flips out of the helix. These are classic features of the S-turn. The guanidine-I riboswitch also possesses a cross-strand purine stack, a characteristic backbone kink on the opposite strand from the S-turn, and stabilizing hydrogen bonds, all of which were first observed in the S-turn of the conserved sarcin-ricin loop in the 23S rRNA (*Correll et al., 1999*). In the PRPP riboswitch, a similar S-turn motif exists at the equivalent position (*Figure 6B*). Equivalent to G89 in guanidine-I, G96 flips out and base pairs with C75 while also hydrogen bonding to PRPP. Notably, the cross-strand purine stack is absent in the PRPP riboswitch, but other S-turn characteristics are preserved. Conversely, the S-turn motif is abolished in the G96A mutant, and no contacts are observed between A96 and other nucleotides. Even more significantly, G95 does not possess the reverse ribose orientation that defines an S-turn. Rather, this region resembles a standard A-form helix with a single nucleotide bulge. The guanine of ppGpp replaces the flipped out guanosine of the former S-turn motif (G89/G96) (*Figure 6*), revealing that the S-turn is a key center of functional plasticity in the *ykkC* RNAs.

## Discussion

Taken together, the present structural and biochemical data shed light on the evolvability of RNA as a whole and of the *ykkC* motif in particular. Just as residue C49 was previously used to distinguish guanidine aptamers from subtype 2 *ykkC* RNAs, here we show that G96 is the residue that differentiates PRPP and ppGpp aptamers. Clearly, the sequence space of the *ykkC* motif is rugged with potential functionality. The existence of *ykkC* RNAs with other gene contexts and unknown ligand specificity further reinforces the diversity of functions that this single RNA structural motif achieves with very small variations in consensus sequence (*Nelson et al., 2017*). Three-dimensional structural models of the wild type and G96A mutant aptamers reveal that the mechanism of specificity switching is recruitment of C75 as a primary effector of ligand recognition (*Figure 6A*). The presence or absence of the S-turn motif governs whether an RNA base or the ppGpp base can pair with C75, and therefore controls the specificity of the aptamer.

The wild-type aptamer featured in the current study binds PRPP at a location very near, but distinct from the binding pocket of the guanidine-I riboswitch. The P0 region, which is not present in the guanidine-I riboswitch, recognizes a portion of the larger ligand; metal ion M3 binds in the location where its parent motif binds guanidine (*Figure 6C*; see also *Figure 3A*). In the S-turn of the sarcin-ricin loop, the bulged G re-inserts into its helix to form a base triple. In an overlay of the S-turns of the sarcin-ricin loop, the guanidine-I riboswitch, and the PRPP riboswitch, the guanidino group of the bulged guanosine in the sarcin-ricin loop overlays almost exactly with the guanidinium cation, and both roughly overlay with metal M3 in the PRPP riboswitch. The common binding site of M3 in the PRPP riboswitch and guanidine in the guanidine riboswitch may be a case of molecular exaptation (the co-option of an existing feature for a new purpose). This is similar to a case documented in a ribozyme created by SELEX, suggesting that structured RNAs are functionally versatile and can readily adapt to new selection pressures (*Lau et al., 2017*). However, the evolutionary relationship of these two aptamers remains uncertain.

The present structural data shed additional light on the potential mechanism of switching in tandem guanine-PRPP aptamers (*Sherlock et al., 2018b*). The PRPP riboswitch (an ON switch), is often found immediately downstream of a guanine riboswitch (an OFF switch), in an IMPLY two-input logic gate (*Figure 6—figure supplement 2*). In these tandem systems, transcription proceeds in all cases except when guanine is present and PRPP is not. This suggests that PRPP binding disrupts formation of the guanine aptamer, allowing transcription to proceed when both ligands are present (*Figure 6—figure supplement 2D*). The *T. mathranii* PRPP aptamer studied in the present work is part of one of these tandem aptamer systems. Its predicted secondary structure shows that formation of the P0 stem of the PRPP aptamer and the P1 stem of the guanine aptamer are mutually exclusive. The present data reveal that the 5′ tail of the PRPP aptamer participates in P0 and plays a central role in

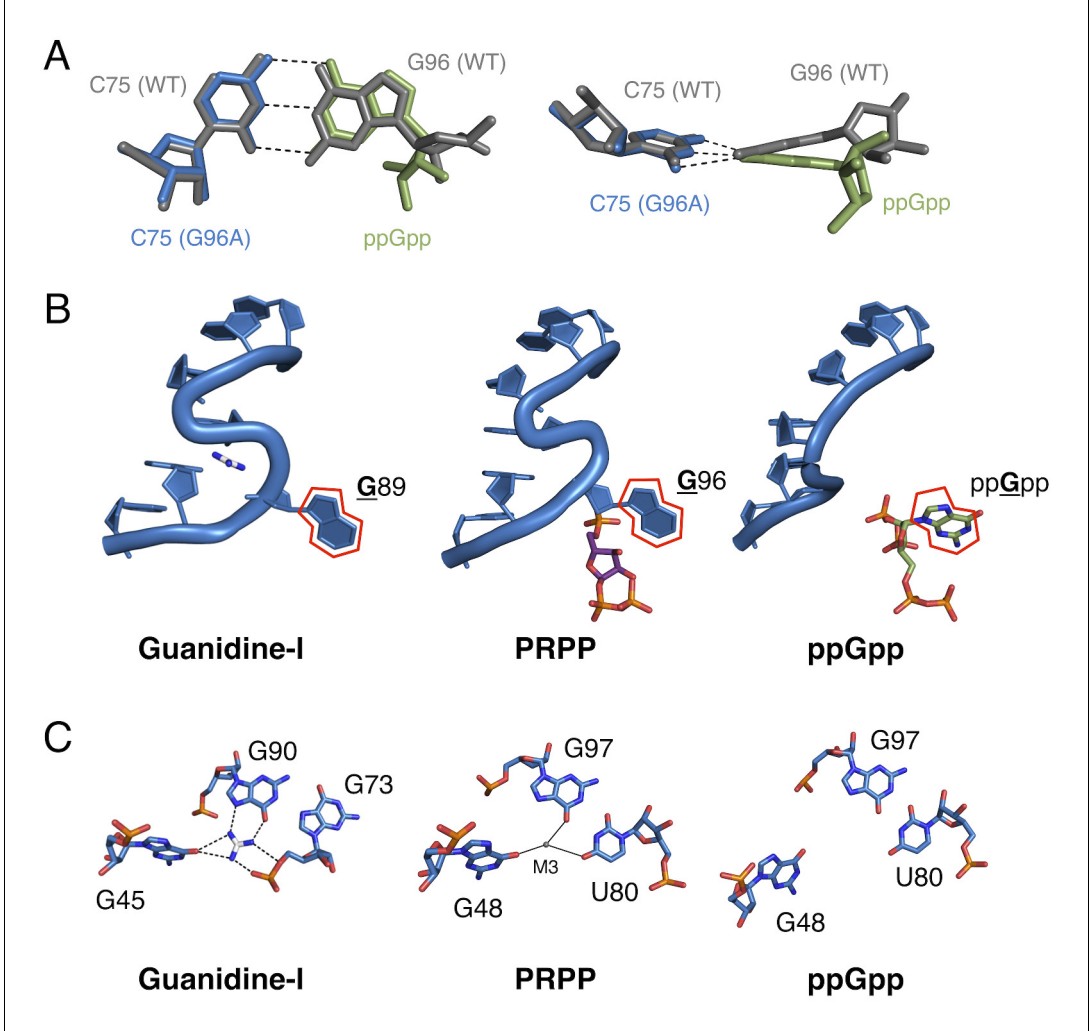

**Figure 6.** Comparison of the guanidine binding site in the guanidine aptamer, the M3 binding site in the PRPP aptamer, and the corresponding site in the G96A mutant. (**A**) Watson-Crick base pairs with C75 in the wild-type and G96A aptamers. Wild-type and G96A aptamer structures are overlaid. Wild-type RNA is shown as gray sticks. G96A RNA is shown as blue sticks. The base and ribose of ppGpp are shown as green sticks. Hydrogen bonds are shown as dashed lines. Left: face-on view of the preserved base pair. Right: edge-on view of the same interaction. (**B**) The S-turn motif in the guanidine and PRPP aptamers, and the equivalent position in the ppGpp aptamer. The RNA is depicted as a blue cartoon. Guanidine is colored by element with white carbons. PRPP is colored by element with purple carbons. ppGpp is colored by element with green carbons. A red outline showcases the position of a conserved guanine base in all three RNA elements. Relative to *Figure 1*, the structure is rotated 180° about the y axis. (**C**) Nucleotides in the guanidine or M3 binding site, or the equivalent site in the ppGpp aptamer. The RNA is colored by element with blue carbons. Guanidine is colored by element with white carbons. Black dashed lines indicate hydrogen bonds. Solid black lines indicate coordination to a metal ion. Individual nucleotides, guanidine and M3 are labeled. Chain A of the PRPP and ppGpp aptamers is shown.

DOI: https://doi.org/10.7554/eLife.36381.015

The following figure supplements are available for figure 6:

**Figure supplement 1.** Conserved interactions between P1 and P3.
DOI: https://doi.org/10.7554/eLife.36381.016

**Figure supplement 2.** Predicted model of switching in the tandem guanine-PRPP riboswitch from T. mathranii.
DOI: https://doi.org/10.7554/eLife.36381.017

**Figure supplement 3.** ppGpp purification and validation.
DOI: https://doi.org/10.7554/eLife.36381.018

PRPP recognition. In the proposed model, PRPP binding stabilizes P0 and disrupts the P1 helix of the guanine aptamer. The IMPLY character of this two-input gate may depend on the relative stabilities of the two helices, which in turn suggests that alternative logic gates could be constructed through mutation of P0 or P1. The ppGpp and T-box riboswitches are also often found in tandem. In contrast with the PRPP/guanine tandem system, the ppGpp and T-box riboswitches each maintain their own expression platform, suggesting that they fold independently. This is consistent with the AND behavior of this logic gate and the unimportance of the order of the two aptamers within the molecular circuit, but future studies in vivo are needed to confirm this.

The observed ability of the *ykkC* scaffold to reach new ligand specificities via mutation of a few key residues is reminiscent of other accounts of adaptability in both proteins and RNAs. In a clinically relevant contemporary example, β-lactamases evolve to expand their catalytic repertoire through mutations in a flexible loop. These mutations preserve the overall architecture of the protein while enabling it to metabolize new variations on a common antibiotic scaffold, contributing to the worldwide threat of antibiotic resistance (*Banerjee et al., 1998*; *Hujer et al., 2001*; *Kurokawa et al., 2000*; *Wachino et al., 2004*). The repurposing of protein scaffolds for the development of new catalysts has also been exploited in the design of novel enzymes including a Diels-Alderase (*Siegel et al., 2010*). In a related RNA example, the *Tetrahymena* ribozyme scaffold supports catalytic activities including self-cleavage, RNA polymerization, and peptide bond hydrolysis, though this is likely due to the placement of the substrates into physical proximity with each other (*Kruger et al., 1982*; *Lau and Ferré-D'Amaré, 2016*; *Piccirilli et al., 1992*; *Zaug and Cech, 1986*). Natural riboswitch aptamers subjected to directed evolution switch specificity, but maintain their overall fold (*Porter et al., 2017*). The existence of riboswitch variants that use the same scaffold but bind slightly different ligands, including the adenine/guanine and cyclic-di-GMP/cyclic-GMP-AMP riboswitch classes, has previously hinted at the adaptability of RNA elements (*Kellenberger et al., 2015*; *Mandal et al., 2003*; *Mandal and Breaker, 2004*; *Ren et al., 2015*; *Serganov et al., 2004*; *Smith et al., 2009*; *Sudarsan et al., 2008*). However, the present finding that a single conservative point mutation in the PRPP aptamer can dramatically alter both ligand specificity and tertiary structure reveals a striking example of RNA plasticity and speaks to the macromolecular evolvability of RNA.

A final key observation in this study is the direct visualization that an RNA element has evolved to specifically recognize PRPP. PRPP is a central metabolite, and likely has played that role since before the metabolic pathways of life's last universal common ancestor (LUCA) were fully developed (*Glansdorff et al., 2008*). It is possible that PRPP was used for the synthesis of nucleotide precursors on the prebiotic Earth (*Akouche et al., 2017*). The finding that an extant RNA specifically recognizes PRPP lends credence to the hypothesis that RNA elements may have been capable of recognizing PRPP before the advent of coded protein synthesis.

## Materials and methods

**Key resources table**

| Reagent type (species) or resource | Designation | Source or reference | Identifiers |
|---|---|---|---|
| Chemical compound, drug | ppGpp standard | TriLink BioTechnologies | TriLink Biotechnologies: N-6001 |
| Chemical compound, drug | PRPP | Millipore Sigma | Millipore Sigma:P8296-25MG |
| Commercial assay or kit | Hampton Research Natrix HT Screen | Hampton Research | Hampton Research:HR2-131 |
| Commercial assay or kit | Harvard Apparatus cassettes | Harvard Apparatus | Harvard Apparatus:742203 |
| Peptide, recombinant protein | Ribose phosphate pyrophosphokinase | Abbexa | Abbexa:abx072019 |
| Sequence-based reagent | T. mathranii genome | NCBI | NCBI:NC_014209.1 |

*Continued on next page*

*Continued*

| Reagent type (species) or resource | Designation | Source or reference | Identifiers |
|---|---|---|---|
| Software, algorithm | CCP4 | https://doi.org/10.1107/S0907444910045749 | CCP4:7.0.042; RRID:SCR_007255 |
| Software, algorithm | Coot | https://doi.org/10.1107/S0907444904019158 | Coot:0.8.6.1; RRID:SCR_014222 |
| Software, algorithm | GraphPad | GraphPad | GraphPad:7.0a; RRID:SCR_002798 |
| Software, algorithm | HKL2000 | https://doi.org/10.1016/S0076-6879(97)76066-X | HKL2000:v0.98.714; RRID:SCR_015547 |
| Software, algorithm | Open Source PyMol | SourceForge https://sourceforge.net/projects/pymol/ | PyMol:v1.8.x; RRID:SCR_000305 |

## RNA transcription and purification

RNA was prepared essentially as in Reiss et al (*Reiss et al., 2017*). Plasmids containing *ykkC* PRPP riboswitch DNA from *T. mathranii* downstream of the T7 promoter were obtained from GeneArt at Thermo Fisher Scientific. The aptamer domain was extended at the 5′ end by one nucleotide to aid transcription by T7 polymerase (*Salvail-Lacoste et al., 2013*). Plasmid DNA was prepared using a QIAgen MaxiPrep kit and the accuracy of the sequence was verified using Sanger sequencing (*Sanger et al., 1977*). Template DNA for transcription was made using PCR with Phusion polymerase and primers 5′-TAATACGACTCACTATAGTGAAAGTGTACC-3′ and 5′-TACGAGTGAAACCTATCC TCCCG-3′. G96A transcription template was generated using the primers 5′-TAATACGACTCACTA TAGTGAAAGTGTACC-3′ and 5′-TACGAGTGAAACCTATCCTCTCGGGCTTTTGTCC-3′. Template was purified using the Zymo Research DNA Clean and Concentrator 500 kit.

RNA was transcribed from 20 ng/μL PCR template using T7 polymerase in the presence of 80 mM HEPES-Na pH 7.5, 5 mM DTT, 1 mM spermidine, 0.12 mg/mL bovine serum albumin, 6 mM NTPs, 44 mM $MgCl_2$, and 1 U/nL inorganic pyrophosphatase (*Hartmann, 2009*). Transcription reactions proceeded for approximately 4 hr at 37°C. Monomeric RNA was exchanged into gel filtration buffer (50 mM MES pH 6.2–6.3, 100 mM KCl, 10 mM $MgCl_2$), filtered, and purified natively on a HiLoad 26/600 Superdex 75 pg gel filtration column in a cold room (6 ± 2°C). Monomers eluted at *ca*. 0.6 column volumes, and were pooled and concentrated to >100 μM.

## Crystallization and structure determination of the wild-type and mutant aptamers

Crystals were grown using the microbatch-under-oil method with 2:1 paraffin:silicon oil. In all cases, crystals appeared within two days. Initial crystallization screening was performed using Hampton Research Natrix HT at 23 and 30°C. To produce the wild type crystals used for data collection, 2 μL of 150 μM RNA in 10 mM $MgCl_2$, 10 mM KCl, 10 mM HEPES-KOH pH 7.5, and 10 mM PRPP (Millipore Sigma) was mixed with 1 μL of a solution of 80 mM sodium chloride, 20 mM barium chloride dihydrate, 40 mM sodium cacodylate trihydrate pH 5.6, 45% v/v (+/-)−2-methyl-2,4-pentanediol (MPD), and 12 mM spermine tetrahydrochloride and incubated at 30°C. To produce the G96A crystals, 150 μM RNA in 10 mM $MgCl_2$, 10 mM KCl, 10 mM HEPES-KOH pH 7.5, and 1 mM ppGpp was mixed with a solution of 80 mM sodium chloride, 40 mM sodium cacodylate pH 7.0, 30% MPD, and 12 mM spermine (1 μL RNA solution plus 0.8 μL reagent) and incubated at 23°C.

Crystals were flash-frozen without further preparation. For the wild type aptamer, a solution was generated using molecular replacement with the *ykkC* guanidine riboswitch as an initial model (PDB ID: 5T83) (*Reiss et al., 2017*). For the G96A mutant, a solution was generated using molecular replacement with chain A of the PRPP riboswitch structure presented in this study as an initial model. Data were processed using HKL-2000 (*Otwinowski and Minor, 1997*). Model building was performed in Coot (*Emsley and Cowtan, 2004*).

The wild type aptamer crystallized in space group *P*2$_1$ with two molecules present in the asymmetric unit. Discussion is for the most part limited to chain A as there is better structural information for this entity than for chain B. The first component modeled was the RNA. Further unaccounted-for electron density was assigned to metal ions and water molecules. This process yielded a structure in

which one significant area of electron density in each chain was unaccounted for. One molecule of PRPP and its two associated metal ions fit well in this area of density.

The G96A aptamer crystallized in space group P1 with four molecules in the asymmetric unit. Discussion in the manuscript is limited to chains A-C, due to chain D yielding generally poorer density. Overall, chain D is consistent with chains A-C, but more subject to error in individual atom positions. Regions disagreeing with the wild type (mainly in the S-turn) were deleted and re-modeled. Very large σ peaks in the difference Fourier map, corresponding to the very electron dense pyrophosphate moieties of ppGpp, were used to identify the ppGpp binding pocket.

Refinement of the two structures was performed with Refmac and Phenix (*Adams et al., 2010*; *Winn et al., 2011*). Refinement was concluded when no more entities could be modeled into the electron density and computational refinement ceased to produce improvements in $R_{work}$ and $R_{free}$. Metal ions were identified by first modeling a magnesium ion and evaluating coordination geometry, B factors, and unaccounted-for density using difference Fourier methods, followed by reassignment where appropriate. The figures of the crystal structure were made in PyMOL (Schrödinger, n.d.). The ligand interaction map was made in ChemDraw.

## Synthesis of (β-$^{33}$P) PRPP and determination of the dissociation constant of the PRPP-aptamer complex

(β-$^{33}$P) PRPP was synthesized using *E. coli* ribose-phosphate pyrophosphokinase (RPPK) obtained from Abbexa. 17.7 µg/mL RPPK was incubated at 37°C for two hours in the presence of 50 mM potassium phosphate dibasic pH 8, 10 mM ribose 5-phosphate, 5 mM MgCl$_2$, and trace quantities of (γ-$^{33}$P) ATP essentially as in Switzer and Gibson (*Switzer and Gibson, 1978*). PRPP and ATP were separated on a native 20% acrylamide gel at 4°C. PRPP was distinguished from the substrate by its faster rate of migration and eluted overnight in 400 µL dH$_2$O at 4°C.

## Purification of SAS1 for synthesis of ppGpp

SAS1 enzyme was expressed and purified based on the protocol in Steinchen et al. (*Steinchen et al., 2015*). Briefly, the SAS1 protein from *B. subtilis* was amplified by colony PCR, cloned into a pET-28aM vector, and transformed into *E. coli* BL21(DE3) cells. A 15 mL starter culture was used to inoculate 1.5 L Terrific Broth plus 50 µg/mL kanamycin and grown at 37°C. At OD$_{600}$ ~0.8, expression was induced with 0.5 mM IPTG and the culture was shaken overnight at 18°C. Cells were then pelleted and lysed using a microfluidizer (lysis buffer: 50 mM Tris, pH 8.0, 300 mM NaCl, 20 mM imidazole, 20 mM MgCl$_2$, 20 mM KCl) and the lysate was run on a nickel column. The protein was eluted from the column with 400 mM imidazole (elution buffer: 50 mM Tris, pH 8.0, 400 mM NaCl, 400 mM imidazole, 0.2 mM TCEP). A band running between 25 and 30 kDa was seen on an SDS-PAGE gel, indicating SAS1 was successfully eluted. The eluted protein was diluted in 50 mM Tris, pH 8.0 and run on a Q column (HiTrap Q column, 5 mL) to remove contaminants. Finally, the Q column fractions were pooled and run on a gel filtration column (Superdex 200, running buffer: 20 mM HEPES-Na, pH 7.5, 200 mM NaCl, 20 mM KCl, 20 mM MgCl$_2$), and a peak eluted consistent with the relevant tetrameric assembly of the protein. The protein was concentrated and frozen at −80°C in aliquots for storage.

## Synthesis and purification of ppGpp

The SAS1 protein accepts GDP (or GTP) and ATP as substrates and catalyzes the transfer of the β and γ phosphates from ATP onto the 3′ end of GDP or GTP to form ppGpp or pppGpp, respectively. To make unlabeled ppGpp for crystallography, a reaction setup based on the protocol of Steinchen et al (*Steinchen et al., 2015*) was used. Briefly, 5 mM GDP, 5 mM ATP, and 5 µM SAS1 were combined in reaction buffer (100 mM HEPES-Na, pH 7.5, 200 mM NaCl, 20 mM MgCl$_2$, and 20 mM KCl) and incubated at 37°C for two hours. A chloroform extraction was performed to remove SAS1, followed by 10-fold dilution in ddH$_2$O and purification by Q column (HiTrap Q HP, 5 mL column volume), where buffer A (10 mM HEPES-KOH, pH 7.5) was used to bind nucleotides to the column and a gradient of buffer B (2 M NaCl) was used to elute the nucleotides. Nucleotides eluted from the column such that the number of phosphates positively correlated with %B. ppGpp eluted last at ~15% buffer B (approximately 300 mM NaCl) (*Figure 6—figure supplement 3*). ppGpp was then precipitated by lithium chloride (LiCl) precipitation. Eluate from the Q column was brought to 1

M LiCl, 4 volumes of ethanol were added, and the tubes were frozen at −20°C before centrifuging at 6000 rpm in an Eppendorf F-45-18-11 fixed-angle centrifuge rotor at 4°C for 10 min to pellet the precipitate. The supernatant was discarded and the pellet was washed twice with cold (−20°C) ethanol, repeating the freezing and pelleting steps between each wash step. After pouring off the ethanol of the final wash, pellets were completely dried. A dry, white powder resulted. Concentration was calculated by measuring UV absorbance at 252 nm ($\varepsilon_{252}$ = 13600 L mol$^{-1}$ cm$^{-1}$).

## Synthesis and purification of (3′-β-$^{32}$P)-ppGpp

A reaction mixture resembling that in the previous section was made, substituting 5 mM ATP for 150 μCi [γ-$^{32}$P]-ATP (Perkin Elmer). (3′-β-$^{32}$P)-ppGpp was purified using a 20% denaturing polyacrylamide gel to separate it from [γ-$^{32}$P]-ATP. The band was soaked in 300 μL ddH$_2$O overnight at 4°C. The gel slice was then filtered off and the solution containing (3′-β-$^{32}$P)-ppGpp was frozen at −20°C for use in binding assays.

## Determination of dissociation constants by equilibrium dialysis

The dissociation constants of the PRPP-RNA and ppGpp-RNA complexes were determined by equilibrium dialysis using cassettes with a 10 kDa cutoff obtained from Harvard Apparatus, essentially as in Reiss et al (Reiss et al., 2017). Trace quantities of radiolabeled ligand were dissolved in equilibrium dialysis buffer (50 mM HEPES-KOH, pH 7.5, 200 mM KCl, 20 mM MgCl$_2$) and were added to one side of the cassette, while varying concentrations of RNA dissolved in the same buffer were added to the other side of the cassette. The cassettes were incubated at room temperature overnight with gentle shaking and recovered by centrifugation. For ppGpp, which experiences negligible amounts of degradation overnight, 20 μL of the recovered material was directly subjected to scintillation counting. For PRPP, 10 μL of the recovered material was subjected to scintillation counting, while another 10 μL was electrophoresed on a denaturing 20% acrylamide gel containing 7.5 M urea. The latter step allowed determination of the amount of PRPP remaining in each sample after overnight incubation at room temperature in the presence of magnesium. The fraction of ligand bound in each cassette was determined using the following equation:

$$F = \frac{(CPM_R * P_R) - (CPM_L * P_L)}{CPM_R * P_R}$$

Where F is the fraction of PRPP bound. $CPM_L$ and $CPM_R$ are the counts per minute measured via scintillation counting of the ligand and RNA sides of the cassette, respectively. $P_L$ and $P_R$ are the percentages of intact PRPP remaining as determined by gel electrophoresis for the ligand and RNA sides of the cassette, respectively. Fitting was performed in GraphPad Prism using the following equation:

$$F = \frac{F_{max} * [RNA]}{K_d + [RNA]}$$

Where F is the fraction of PRPP bound, $F_{max}$ is the maximum fraction of PRPP bound, [RNA] is the concentration of RNA, and $K_d$ is the dissociation constant.

All binding data consist of three technical replicates, of which the arithmetic mean and standard deviation are represented in Figure 1—figure supplement 2. All replicates were performed using a single stock of RNA from the same round of in vitro transcription and purification. Each data point in each replicate was collected using a different equilibrium dialysis cassette with independently diluted RNA solutions. Data were fit to a single-binding hyperbolic curve, with $B_{max}$ floating or constrained as follows. TmaWT binding to PRPP and TmaG96A binding to ppGpp fully reached saturation and $B_{max}$ was allowed to float. These $B_{max}$ values fit here should represent the fraction of radiolabeled ligand available for binding. TmaWT binding to ppGpp and TmaG96A binding to PRPP did not reach saturation, so the $B_{max}$ was constrained to equal the $B_{max}$ values for TmaG96A binding to ppGpp and TmaWT binding to PRPP, respectively. Values shown in Supplementary file 1 are the dissociation constant ($K_d$) from the hyperbolic fit plus or minus the standard error of the fit calculated by GraphPad Prism.

## Secondary structure prediction

Secondary structure predictions were obtained from mFold using the default settings (*Zuker, 2003*). NCBI Reference Sequence: NC_014209.1, location 657606 to 657788 was used to predict the secondary structure of the aptamers in the presence of guanine and absence of PRPP. Location 657617 to 657800 was used to predict the secondary structures of the aptamers in the absence of guanine, in the presence of PRPP, and in the presence of both ligands. The secondary structure of the transcription terminator was predicted using location 657606 to 657841. The secondary structure deemed most likely by the program was used for interpretation. Bound guanine aptamer and bound PRPP aptamer secondary structure predictions are consistent with X-ray crystallography data.

## Accession numbers

Coordinates have been deposited in the Protein Data Bank (PDB) with accession numbers 6CK5 for the wild-type PRPP aptamer and 6CK4 for the G96A ppGpp aptamer.

## Acknowledgements

The authors thank the synchrotron beamline staff at the Northeastern Collaborative Access Team (NE-CAT) at the Advanced Photon Source (APS) for their assistance; Jimin Wang, who provided important help with structure determination, and Michael Strickler from the Yale Center for Structural Biology; Maddie Sherlock and Ron Breaker for providing data on the *ykkC* variant subtypes in advance of publication and helpful discussion; and David Hiller for comments and help with intellectual development of the manuscript. The authors acknowledge Brady Summers and Yong Xiong for assistance with purification of SAS1 protein. This work was funded by NIH grant GM022778 to SAS.

## Additional information

### Funding

| Funder | Grant reference number | Author |
|---|---|---|
| National Institutes of Health | GM022778 | Scott A Strobel |

The funders had no role in study design, data collection and interpretation, or the decision to submit the work for publication.

### Author contributions

Andrew John Knappenberger, Conceptualization, Formal analysis, Validation, Investigation, Visualization, Methodology, Writing—original draft, Writing—review and editing, Determined the crystal structure of the wild-type aptamer and its affinity for PRPP; Caroline Wetherington Reiss, Conceptualization, Formal analysis, Validation, Investigation, Visualization, Methodology, Writing—original draft, Writing—review and editing, Determined the crystal structure of the G96A mutant, its affinity for PRPP and ppGpp, and the affinity of the wild-type aptamer for ppGpp; Scott A Strobel, Conceptualization, Resources, Supervision, Funding acquisition, Project administration, Writing—review and editing

### Author ORCIDs

Andrew John Knappenberger (iD) http://orcid.org/0000-0003-2659-4305
Caroline Wetherington Reiss (iD) http://orcid.org/0000-0002-6385-1879
Scott A Strobel (iD) http://orcid.org/0000-0001-8402-4226

### Decision letter and Author response
Decision letter https://doi.org/10.7554/eLife.36381.029
Author response https://doi.org/10.7554/eLife.36381.030

# Additional files

## Supplementary files

• Supplementary file 1. X-ray crystallography statistics from data collection and refinement.
DOI: https://doi.org/10.7554/eLife.36381.019

• Transparent reporting form
DOI: https://doi.org/10.7554/eLife.36381.020

• Reporting file 1
DOI: https://doi.org/10.7554/eLife.36381.021

## Data availability

Diffraction data and coordinates have been deposited under the accession codes 6CK4 and 6CK5.

The following datasets were generated:

| Author(s) | Year | Dataset title | Dataset URL | Database, license, and accessibility information |
|---|---|---|---|---|
| Knappenberger AJ, Reiss CW, Strobel SA | 2018 | PRPP riboswitch from T. mathranii bound to PRPP | https://www.rcsb.org/structure/6CK5 | Publicly available at the RCSB Protein Data Bank (accession no: 6CK5) |
| Reiss CW, Knappenberger AJ, Strobel SA | 2018 | G96A mutant of the PRPP riboswitch from T. mathranii bound to ppGpp | https://www.rcsb.org/structure/6CK4 | Publicly available at the RCSB Protein Data Bank (accession no: 6CK4) |

The following previously published dataset was used:

| Author(s) | Year | Dataset title | Dataset URL | Database, license, and accessibility information |
|---|---|---|---|---|
| Reiss CW, Xiong Y, Strobel SA | 2018 | Structure of a guanidine-I riboswitch from S. acidophilus | https://www.rcsb.org/structure/5T83 | Publicly available at the RCSB Protein Data Bank (accession no: 5T83) |

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
