## [Decision Letter]

Thank you for submitting your article "Structures of two aptamers with differing ligand specificity reveal ruggedness in the functional landscape of RNA" for consideration by *eLife*. Your article has been reviewed by four peer reviewers, and the evaluation has been overseen by Timothy Nilsen as Reviewing Editor and James Manley as the Senior Editor. The reviewers have opted to remain anonymous.

The reviewers have discussed the reviews with one another and the Reviewing Editor has drafted this decision to help you prepare a revised submission.

Summary:

The manuscript by Knappenberger et al. describes the exciting structure determination and analysis of the PRPP riboswitch aptamer domain, and a single nucleotide mutant that imparts effector specificity for ppGpp. This remarkable change in specificity is even more interesting because the wild-type aptamers are related closely to the recently described class of guanidium-I-sensing riboswitches derived from the ykkC superfamily.

The reviewers agreed that the work was important and in principle appropriate for *eLife*. However, each reviewer raised a number of potentially significant issues regarding the interpretation of the crystallographic data. These include the positioning of and stability of PRPP as well as the assignment of metal ion binding sites. Please address all of these concerns as thoroughly as possible. In this case any revised manuscript will require re-review by the same referees. Additionally we assume that the two Sherlock papers will be In Press if your revised manuscript is accepted.

*Reviewer #1:*

The manuscript by Knappenberger et al. describes the exciting structure determination and analysis of the PRPP riboswitch aptamer domain, and a single nucleotide mutant that imparts effector specificity for ppGpp. This remarkable change in specificity (reported by the authors as a~40,000-fold change) is even more interesting because the wild-type aptamers are related closely to the recently described class of guanidium-I-sensing riboswitches derived from the ykkC superfamily. Crystal structures reveal that the PRPP sensor and mutant bind their ligands in a four-helix junction that utilizes a novel P0 helix – not found in the guanidium-I riboswitch – to impart new ligand binding properties. The added P0 structural element has implications for the gene regulatory mechanisms of tandem guanine-PRPP riboswitch sensors that work synergistically to compose a dual input "imply" gate that shares an expression platform. Details of ligand binding are described including the association of metal ions. An interesting aspect of sensing is the existence of an S-turn motif that is integral to guanidinium or PRPP sensing. However, extrusion of a key S-turn guanine obstructs the base in ppGpp from binding. The latter block necessitates an S-turn conformational adjustment caused by a single G96A point mutant that imparts ppGpp effector recognition over PRPP.

Overall, the work is very interesting and should be of broad interest to the journal audience. However, there are some technical details regarding the assignment of the ions and orientation of the PPRP effector that should be addressed to provide greater confidence in the crystallographic model. Another point of concern is citing unpublished or "in press" work, which may not agree with the journal policy.

1) Introduction, second paragraph and throughout the manuscript, the authors make reference to Sherlock et al. in press as two separate manuscripts. Please provide the "Sherlock et al. in press" manuscripts with more complete information (i.e., all authors, title, journal). Since there are two manuscripts, they need to be distinguished as 2018a or 2018b, or by the appropriate journal style.

2) Introduction, fourth and fifth paragraphs, and elsewhere. The authors refer to "Sherlock et al., unpublished." This may not conform the journal guidelines for citations. The reviewer objects to citing unpublished work because it is not a rigorous source of information that is transparent for the journal readership.

3) Supplementary file 1. The reviewer has a number of suggestions to improve the table.

First, the unit cell cannot be known to an accuracy of 1/1000th Å based on the mode of data collection. HKL2000 will report the errors and these could be indicated or the cell should be reported to 1/10th of an Ångstrom. The same is true of the angles.

Second, the reviewer recommends using R_p.i.m._ instead of *Rmerge*. The former accounts for redundancy in the data collection and should replace the latter (see Karplus and Diedrichs (Curr Opin Struct Biol 34, 60) and Weiss (J. Appl. Cryst. 34, 130-135). This metric is also provided by HKL2000.

4) Figure 2. The mode of metal ion coordination is unclear based on the drawing. To clarify the scheme, please provide the name of the nucleotide atom in panel B that forms an interaction with PRPP. The underlying reason for this request is that the ligand could be modeled in a different manner in the electron density.

5) One aspect of the presentation that could be improved is to provide greater evidence for the orientation of modeled PRPP. Rotation of PRPP by 180° about the y-axis would place the pyrophosphate into M2. The reviewer noted this possibility prior to reading the 180° rotation of ppGpp in the ensuing section. In this respect, more should be done to explore this possibility as a supplemental figure. For example, compare the SA omit electron density for both orientations (or use averaged kicked maps or composite omit rebuild maps, which are likely the best – but slowest to run; use 2Fo-Fc coefficients). How do the R_free_ values compare for each orientation? How do the real space correlation coefficients compare?

6) It is unclear what metal ions are present in sites M1, M2 and M3. These are stated to be consistent with Ba^2+^ (subsection “The structure of the wild-type PRPP aptamer and a single point mutant ppGpp aptamer”, seventh paragraph) but actual distance information and geometry should be provided. Why? It seems unusual that Ba^2+^ would bind in place of Mg^2+^ since the latter has significantly shorter coordination distances and distinct octahedral geometry. Mg^2+^ is the expected intracellular ion of course. Anomalous scattering from Ba^2+^ might help to more definitively model the ions, which is still difficult to define at this resolution. In these respects, the coordination distances between the ions and the ligands should be mentioned. (the aforementioned paragraph notes the coordination of M2 is consistent within inner sphere coordination. Please be more specific). What are the B-factors of the Ba^2+^ ions? If they are modeled as Mg^2+^, are they unreasonably low? If another contour level is add to the maps, is the s-to-n consistent with Ba^2+^ (e.g., 10 σ or greater peaks).

7) The observation of Ba^2+^ in the complex with PRPP suggests that the ion should be sufficient for binding in equilibrium dialysis experiments. The experimental methods describe the use of Mg^2+^ (subsection “Determination of dissociation constants by equilibrium dialysis”, first paragraph) Was Ba^2+^ tried? If Ba^2+^ showed binding, this could be a stronger case for the observation of the assignment of Ba^2+^ in the electron density maps. The reviewer realizes that these are not easy experiments, so analysis with Ba^2+^ may not be feasible. Why not use isothermal titration calorimetry?

*Reviewer #2:*

The study by Knappenberger et al. characterizes the structure of a subtype of the guanidine-I riboswitch that binds the nucleotide precursor PRPP. The study goes even further to describe the structure of a point mutant that switches specificity to the bacterial alarmone ppGpp. By detailing how small nucleotide changes can alter the specificity of riboswitch variants, the authors show how structured RNAs can readily evolve new functions. The descriptions of ligand recognition by each riboswitch is very thoroughly and the structure guided rationalization of altered ligand specificity between guanidine-I, PRPP, and ppGpp riboswitches is strong. Overall, the study is significant for describing ligand recognition by two newly characterized riboswitches and for looking at the evolution of riboswitch variants through a structural lens. The study should be of interest to the broad readership of the journal. I do have a few concerns:

1) The I/sigmaI values reported for each structure are on the low side (1.27 for PRPP and 0.67 for ppGpp) and the difference between R_free_ and R_work_ for each structure is on the high side. Perhaps the structures have been refined to an artificially high resolution. Furthermore, both structures appear to be missing portions of the chain.

2) The authors should cite the paper by Battaglia et al. describing the structure of guanidine-I.

*Reviewer #3:*

This paper describes the structural elucidation of the PRPP riboswitch aptamer (ykkC 2b subtype) and a synthetic ppGpp riboswitch aptamer made by a G96A point mutation of the prior aptamer, as representative of the ykkC 2a subtype. This research group previously had solved the structure of the ykkC 1 subtype (guanidine-I), and there are two main points of interest for this current work: (1) comparison of these two new structures to the guanidine-I structure provides detailed insight into how a riboswitch scaffold can dramatically switch ligand specificity, from a positively charged ion to highly negatively charged compounds; (2) comparison between the two new structures demonstrates how a point mutation leads to full switch in ligand specificity, as both WT and G96A riboswitches are ~50-900 fold selective against the non-cognate ligand. Together these points highlight the functional plasticity of RNA; other examples have been shown before, but none with as dramatic a change in the ligand structure. The authors also state that the PRPP riboswitch provides evidence for the RNA world theory.

Overall, the work is very solid – the structures are to 2.5 and 3.1 Å resolution, so the authors are careful to point out what they cannot conclude due to limitations of resolution, and to present other support for choices made regarding modeled ligand conformations. Each structure presented unique challenges that had to be overcome: PRPP is a highly unstable ligand, whereas the native ppGpp aptamers did not crystallize and led to use of the G96A mutant. There is a strong reliance on unpublished data from Sherlock et al., so these two manuscripts are linked – this reviewer feels that this paper relies on the other to provide more biological context. However, there is definitely insight gained from these structures: specifically, the way that the guanosine base in ppGpp replaces G96 in the PRPP structure is satisfying to see; and surprisingly, how the ligand orientation is completely flipped.

1) Compound identity for synthesized PRPP and ppGpp. This is not a chemistry journal, but how did authors validate compound identity? Because "riboswitch binds synthesized compound, therefore compound identity is X" would be circular reasoning. Is the resolution high enough to serve as validation of ligand identity? Also was wondering if resolution was high enough to show that ligands were not hydrolyzed in the structure, e.g. GMP+PPi instead of GTP is common.

2) – "A common ancestral RNA likely diverged to recognize guanidine, PRPP, and ppGpp in spite of the chemical and structural diversity among these ligands" This line is too speculative. What evidence do you have that one became three, vs. first was guanidine, then branched to two new, or vice versa?

3) Figure 1—figure supplement 1 – "PRPP" control sample, how was this sample treated? It is unclear if it was treated under same reaction conditions, although text suggests PRPP would be totally degraded under those conditions (but not shown?)

4) Figure 2 – Results text state that M3 "forms a water-mediated coordination to the 5-phosphate" but this is not shown in B?

5) Figure 6C – does resolution allow for assignment that there is no ion or water bound in the guanidine/metal pocket shown?

6) Discussion: It should be made clear that this study does not demonstrate that the G96A mutation is sufficient to alter gene regulation in vivo, although the aptamer selectivity is altered. See Mandal and Breaker 2004 for precedence. Along these lines, what is the range of% identity for natural PRPP and ppGPP riboswitches? This would be helpful to know.

---

## [Author Response]

Reviewer #1:

[…] Overall, the work is very interesting and should be of broad interest to the journal audience. However, there are some technical details regarding the assignment of the ions and orientation of the PPRP effector that should be addressed to provide greater confidence in the crystallographic model. Another point of concern is citing unpublished or "in press" work, which may not agree with the journal policy.1) Introduction, second paragraph and throughout the manuscript, the authors make reference to Sherlock et al. in press as two separate manuscripts. Please provide the "Sherlock et al. in press" manuscripts with more complete information (i.e., all authors, title, journal). Since there are two manuscripts, they need to be distinguished as 2018a or 2018b, or by the appropriate journal style.2) Introduction, fourth and fifth paragraphs, and elsewhere. The authors refer to "Sherlock et al., unpublished." This may not conform the journal guidelines for citations. The reviewer objects to citing unpublished work because it is not a rigorous source of information that is transparent for the journal readership.

One significant issue raised by the reviewers and the editors was the status of the work cited from the Breaker lab. We are happy to report that the two Sherlock et al. papers referenced in our work have been accepted by *eLife* and PNAS. This enables us to properly cite their work in this revised manuscript. Sherlock, Sudarsan, and Breaker 2018 is not yet available online as of our re-submission, but it is expected to appear in the very near future.

3) Supplementary file 1. The reviewer has a number of suggestions to improve the table.First, the unit cell cannot be known to an accuracy of 1/1000th Å based on the mode of data collection. HKL2000 will report the errors and these could be indicated or the cell should be reported to 1/10th of an Ångstrom. The same is true of the angles.

Corrected.

Second, the reviewer recommends using R_p.i.m._ instead of Rmerge. The former accounts for redundancy in the data collection and should replace the latter (see Karplus and Diedrichs (Curr Opin Struct Biol 34, 60) and Weiss (J. Appl. Cryst. 34, 130-135). This metric is also provided by HKL2000.

R_p.i.m._ is now included in Supplementary file 1 in addition to *Rmerge*.

4) Figure 2. The mode of metal ion coordination is unclear based on the drawing. To clarify the scheme, please provide the name of the nucleotide atom in panel B that forms an interaction with PRPP. The underlying reason for this request is that the ligand could be modeled in a different manner in the electron density.

New supplemental figure (Figure 1—figure supplement 2) shows coordination distances to the three metals and the text now describes the ambiguities inherent in metal assignment at this resolution.

5) One aspect of the presentation that could be improved is to provide greater evidence for the orientation of modeled PRPP. Rotation of PRPP by 180° about the y-axis would place the pyrophosphate into M2. The reviewer noted this possibility prior to reading the 180° rotation of ppGpp in the ensuing section. In this respect, more should be done to explore this possibility as a supplemental figure. For example, compare the SA omit electron density for both orientations (or use averaged kicked maps or composite omit rebuild maps, which are likely the best – but slowest to run; use 2Fo-Fc coefficients). How do the R_free_ values compare for each orientation? How do the real space correlation coefficients compare?

We modeled PRPP in the alternative orientation suggested by the reviewer. In this orientation, the R_free_ rises from 0.2523 to 0.2537. It results in a short (2 Å) hydrogen bond between PRPP and the N1 atom of G6, and forces two non-bridging oxygens in the pyrophosphate into implausibly close proximity (1.8 Å) with one another. All indications are that this is not the correct orientation of the molecule within the RNA.

6) It is unclear what metal ions are present in sites M1, M2 and M3. These are stated to be consistent with Ba^2+^ (subsection “The structure of the wild-type PRPP aptamer and a single point mutant ppGpp aptamer”, seventh paragraph) but actual distance information and geometry should be provided. Why? It seems unusual that Ba^2+^ would bind in place of Mg^2+^ since the latter has significantly shorter coordination distances and distinct octahedral geometry. Mg^2+^ is the expected intracellular ion of course. Anomalous scattering from Ba^2+^ might help to more definitively model the ions, which is still difficult to define at this resolution. In these respects, the coordination distances between the ions and the ligands should be mentioned. (the aforementioned paragraph notes the coordination of M2 is consistent within inner sphere coordination. Please be more specific). What are the B-factors of the Ba^2+^ ions? If they are modeled as Mg^2+^, are they unreasonably low? If another contour level is add to the maps, is the s-to-n consistent with Ba^2+^ (e.g., 10 σ or greater peaks).7) The observation of Ba^2+^ in the complex with PRPP suggests that the ion should be sufficient for binding in equilibrium dialysis experiments. The experimental methods describe the use of Mg^2+^ (subsection “Determination of dissociation constants by equilibrium dialysis”, first paragraph) Was Ba^2+^ tried? If Ba^2+^ showed binding, this could be a stronger case for the observation of the assignment of Ba^2+^ in the electron density maps. The reviewer realizes that these are not easy experiments, so analysis with Ba^2+^ may not be feasible. Why not use isothermal titration calorimetry?

We thank the reviewer for raising this important point. We have carefully reviewed the metal assignments for M1, M2, and M3 within the PRPP structure. Because there is a mixture of divalent metals in the crystal condition (both Mg^2+^ and Ba^2+^ are present), partial occupancy is a potential complicating factor. Both Mg^2+^ and Ba^2+^ support PRPP binding and do so at similar affinities (see above). The data support the assignment of M1 and M3 as having significant occupancy by Ba^2+^. Assignment of M1 or M3 as magnesium result in very large positive peaks occur in the F_o_-F_c_ map. Upon closer inspection, there is reasonable case to be made for assigning M2 as Mg^2+^. Although coordination distances (~2.8 Å to 3.0 Å) are longer than expected for Mg^2+^, the B factor (64 Å2) and difference map are reasonable. All three sites are likely to be partially occupied by both Mg^2+^ and Ba2+ which complicates the assignments. A sentence to this effect has been added to the text.

Reviewer #2:

[…] 1) The I/sigmaI values reported for each structure are on the low side (1.27 for PRPP and 0.67 for ppGpp) and the difference between R_free_ and R_work_ for each structure is on the high side. Perhaps the structures have been refined to an artificially high resolution.

We used a combination of the CC_1/2_ and the point at which the slope of the Wilson plot approaches zero to assign cutoff points. We used the widely accepted CC_1/2_ value of 0.15 to cut the data.

Furthermore, both structures appear to be missing portions of the chain.

These nucleotides are disordered in the structure so we did not model them.

2) The authors should cite the paper by Battaglia et al. describing the structure of guanidine-I.

We particularly thank the reviewers for catching this omission. The citation has been added.

Reviewer #3:

[…] 1) Compound identity for synthesized PRPP and ppGpp. This is not a chemistry journal, but how did authors validate compound identity? Because "riboswitch binds synthesized compound, therefore compound identity is X" would be circular reasoning. Is the resolution high enough to serve as validation of ligand identity? Also was wondering if resolution was high enough to show that ligands were not hydrolyzed in the structure, e.g. GMP+PPi instead of GTP is common.

PRPP was purchased from Σ Aldrich and validated by the company. ppGpp was made using previously established methods and verified by comparison of anion-exchange retention time with a ppGpp standard (purchased from TriLink BioTechnologies). A supplemental figure has been added to show this validation (Figure 6—figure supplement 3).

2) "A common ancestral RNA likely diverged to recognize guanidine, PRPP, and ppGpp in spite of the chemical and structural diversity among these ligands" This line is too speculative. What evidence do you have that one became three, vs. first was guanidine, then branched to two new, or vice versa?

We did not intend to suggest an order of divergence between these RNA classes. The text has been revised to clarify this point.

3) Figure 1—figure supplement 1 – "PRPP" control sample, how was this sample treated? It is unclear if it was treated under same reaction conditions, although text suggests PRPP would be totally degraded under those conditions (but not shown?)

Explanatory text has been added to the legend of Figure 1—figure supplement 1.

4) Figure 2 – Results text state that M3 "forms a water-mediated coordination to the 5-phosphate" but this is not shown in B?

Added.

5) Figure 6C – does resolution allow for assignment that there is no ion or water bound in the guanidine/metal pocket shown?

We do see ions and water molecules in this area, but they are too distant to be true coordination or hydrogen bonding contacts. Positioning of these species varies among chains in the asymmetric unit.

*6) Discussion: It should be made clear that this study does not demonstrate that the G96A mutation is sufficient to alter gene regulation* in vivo*, although the aptamer selectivity is altered. See Mandal and Breaker 2004 for precedence. Along these lines, what is the range of% identity for natural PRPP and ppGPP riboswitches? This would be helpful to know.*

Done.